# Cognitive load and visual attention assessment using physiological eye tracking measures in multimedia learning

Fatemeh Shahnabati[1], Atefeh Sabourifard[1], S. Hamid Amiri[1], Alireza Bosaghzadeh[1], Reza Ebrahimpour [2,3]*

1 Department of Computer Engineering, Shahid Rajaee Teacher Training University, Tehran, Islamic Republic of Iran, 2 Center for Cognitive Science, Institute for Convergence Science and Technology (ICST), Sharif University of Technology, Tehran, Islamic Republic of Iran, 3 School of Cognitive Sciences (SCS), Institute for Research in Fundamental Sciences (IPM), Niavaran, Tehran, Iran

* ebrahimpour@sharif.edu

## Abstract

Effective multimedia content design can boost performance, capture visual attention, and optimize cognitive load. The current study employs eye-tracking technology to establish metrics to measure cognitive load, analyze visual attention allocation, and evaluate learners' performance in English language learning. The study focuses on creating and comparing two different multimedia presentations. The differentiation between them lies in their adherence to or deviation from Mayer's educational multimedia design principles: coherence, signaling, and spatial contiguity. participants were randomly assigned to two groups. The first group viewed with principles version, while the second group viewed without principles version, during which their eye movement data were collected. Subsequently, both groups participated in a recall test and completed the NASA-TLX questionnaire. The research establishes connections between specific eye-tracking parameters, subjective cognitive load scores, and recall test results through regression models and analyzes fixation distributions. The study also delves into microsaccades rate and changes in pupil size, each analyzed within times of interest. The study's findings indicate that the examined metrics can significantly help distinguish between the two conditions: principles and no principles. These metrics are pertinent for assessing individuals' cognitive load and visual attention and serve as beneficial indicators for gauging the efficacy of the designed multimedia content.

## 1. Introduction

In contrast to traditional educational methods reliant solely on textual information, educational multimedia offers an immersive learning environment that seamlessly integrates visual and verbal cues to enhance the learning experience. For instance,

**Data availability statement:** The data supporting the conclusions of this article is made available without undue reservation in the link below: https://doi.org/10.5281/zenodo.15101629.

**Funding:** This work was supported by the Iranian National Science Foundation (INSF) (under proposal number of 4015666 to Reza Ebrahimpour).

**Competing interests:** The authors have declared that no competing interests exist.

research indicates that university students who combined online instructional videos with traditional classroom teaching achieved superior outcomes compared to those exposed solely to conventional face-to-face instruction through an approach known as blended learning [1]. The effectiveness of educational multimedia is most pronounced when multimedia design and the integration of visual and verbal elements align harmoniously with the functioning of the human brain [2]. Consequently, the impact of educational multimedia can vary significantly based on its design and its influence on learning improvement [2]. Given the limitations of learners' cognitive capacities, employing multimedia design strategies that effectively engage learners with educational content while minimizing the unnecessary cognitive load on working memory becomes paramount, ultimately leading to more meaningful learning experiences [3].

Cognitive load is a multifaceted concept reflecting the cognitive demands imposed by specific tasks on a learner's cognitive system [4]. The Cognitive Load Theory (CLT) categorizes cognitive load into three types: intrinsic, extraneous, and germane. Intrinsic cognitive load, shaped by the interaction between the material and learners' expertise, is influenced by the complexity of material interactivity [5]. This cannot be directly modified or minimized through instructional design. The more the number of elements that are related to each other in the educational materials, the more their complexity will increase; therefore, the intrinsic cognitive load will increase. Extraneous cognitive load results from poorly designed instructions, adding unnecessary burdens beyond intrinsic cognitive load. This additional load hinders learning and should be minimized to optimize cognitive resource allocation [6,7]. Germane cognitive load pertains to processes contributing to schema construction and automation, directly fostering learning. Both extraneous and germane loads are modifiable by instructional designers [6,7]. Educational multimedia must be meticulously crafted to alleviate unnecessary cognitive load, maximize cognitive resource allocation, and enhance working memory capacity, bolstering germane load and improving learning outcomes. Mayer's work encapsulates 12 principles for effective multimedia presentations that mitigate undue cognitive load [8]. Three of these principles include: 1. Coherence Principle, 2. Signaling Principle, 3. Spatial Contiguity Principle. Three primary methodologies are employed to measure cognitive load: subjective rating, performance-based measures, and physiological measures [4,9,10]. Subjective rating methods assume that people can estimate their cognitive processes and report their mental efforts [11]. These methods typically hinge on questionnaires spanning either one-dimensional or multidimensional formats. While unidimensional scales capture overall cognitive load, multidimensional scales encompass diverse components like fatigue, mental effort, and frustration [12]. Notably, the NASA-TLX scale, assessing six dimensions, finds widespread application. Nevertheless, subjective rating methods exhibit limitations, including potential response exaggeration, individual biases, and the inability to track cognitive load fluctuations during testing. Performance-based measurement quantifies cognitive load based on individual task performance or behavior. For instance, the dual-task paradigm involves concurrently executing a secondary task alongside the primary task, with secondary task

performance as a gauge for changes in cognitive load within the primary task. However, while performance-based methods facilitate real-time tracking of cognitive load variations, they often prove better suited to controlled laboratory environments, proving practically challenging or unfeasible for real-world application. Physiological methods, constituting the third category of cognitive load measurement, offer the capability to discern cognitive shifts via specific physiological variables (such as fMRI, electroencephalogram, and heart rate variability) [13–15], showcasing real-time sensitivity and resolution [10]. EEG stands as a prominent tool leveraging brain activity to gauge cognitive load [16], through feature extraction from EEG signals, established a viable technique for assessing cognitive load in educational multimedia [16]. Farkish and colleagues by measuring the cognitive load in educational multimedia using EEG signals, showed that following Mayer's multimedia design principles significantly reduces the cognitive load of users [17]. This underscores the potential of physiological measures to provide nuanced insights into cognitive load dynamics during multimedia engagement.

Eye-tracking technology has emerged as a vital tool across numerous studies for observing cognitive processes and gauging attention in multimedia education experiments [15]. Various aspects of eye movement behavior are categorized into distinct groups, including:

1. **Fixational and Saccadic Measures**: This category encompasses metrics like fixation count, fixation duration, microsaccadic measures, saccade amplitude, saccade velocity, and saccade latency.

2. **Blink Measures**: Blink rate and blink amplitude provide insights into visual engagement.

3. **Visual Search Measures**: Parameters such as time to first fixation, gaze transition, scan path similarity, and dwell time shed light on visual exploration patterns.

4. **Pupil Measurements**: Pupil size variation and The Index of Cognitive Activity (ICA) contribute to understanding cognitive processing.

Research has shown that these criteria offer valuable information about visual information processing, learning strategies, and visual search patterns and can even serve as indicators for assessing cognitive load. For instance, Chen and colleagues found that longer and slower fixations correlate with higher attentional effort demands [18]. The relationship between cognitive load and fixation duration can vary with the task; fixation duration might increase [19] or decrease [20,21] as mental workload intensifies. Similarly, the frequency of saccadic movements depends strongly on the nature of the task at hand [22]. Microsaccades, covering less than 1° of the visual angle, are crucial in preventing currently viewed visual information from fading [23]. Recent studies suggest a link between microsaccade occurrence and cognitive load. According to [24], greater task complexity corresponds to a decrease in microsaccade rate and an increase in microsaccade amplitude. Additionally, [25] revealed that the microsaccadic rate drops under high-load conditions of a memory task compared to low-load conditions. Conversely, other evidence indicates that microsaccadic frequency increases with the visual complexity of a task [26]. This underscores the potential of microsaccades as a window into cognitive processing during various cognitive tasks. The findings of [27] highlight a noteworthy pattern: Individuals tend to maintain their gaze on a specific area of an image or scene for an extended duration to enhance attention and minimize distractions. Consequently, a reduced blink rate emerges as an informative cue indicative of heightened attention and increased concentration on the scene. Another significant gauge linked to visual information processing is the frequency of gaze transitions between Areas of Interest (AOIs). This metric offers insights into whether learners effectively establish connections between different informational components or encounter challenges integrating diverse elements [28]. Diverse interpretations of this metric have emerged across studies [29]. It sometimes benefits cognitive processes such as organization and integration [30,31]. However, in other scenarios, excessive fixation transitions are viewed as detrimental to learning, signaling split attention [32]. The fluctuation of pupil diameter, believed to mirror changes in brain activity and human cognition, represents another fascinating avenue of research. Scholars have examined the correlation between pupil size, cognitive load, and varied tasks, encompassing activities like driving while engaged in conversation, solving mathematical

problems, recalling numbers, and processing visual stimuli [33–35]. This exploration underscores the dynamic relationship between pupil changes and cognitive engagement across diverse cognitive endeavors.

## 2. This study

Two primary objectives guided our study. Firstly, we aimed to uncover the impact of deviating from Mayer's multimedia design principles on the cognitive load experienced by students' working memory. Secondly, we wanted to have a systematic and methodological approach to measure cognitive load and processing and visual resource allocation eye tracking tools. We wanted to know if the eye behaviors differ between the two groups when the cognitive load changes due to watching different multimedia versions.

While numerous investigations have delved into multimedia design principles, firstly, their focus has centered on learning outcomes. We intended to explore the cognitive load resulting from principle violations using physiological eye tracking data. Secondly, within Mayer's 12 proposed principles, most attention in eye-tracking studies has been directed toward modality and signaling. In contrast, certain principles, such as spatial contiguity and coherence, have received relatively limited scrutiny. For instance, a solitary study delved into gaze behavior during multimedia learning to probe the spatial contiguity principle [36]. However, this study failed to establish a significant contiguity effect on learning outcomes like retention or transfer test scores [37]. Furthermore, the effectiveness of video and animation-based training and dynamic content compared to static training content necessitates further investigation [37]. Equally pivotal is the investigation into the impact of adhering to or violating principles of educational multimedia design on eye behaviors. While eye behavioral measures have been applied in diverse studies, spanning mathematical calculations to human-computer interaction question comprehension, comprehending how observing or violating multimedia design principles influences eye behaviors warrants deeper exploration. For instance, while probing microsaccades or pupil responses could serve as avenues for understanding shifts in cognitive load, few studies have systematically addressed alterations in microsaccade-related metrics or pupil size during multimedia engagement. Moreover, the literature underscores contradictory findings regarding pattern changes and the susceptibility of many of these metrics to the task context. Thus, an enhanced understanding of these metrics, particularly within video-oriented multimedia education, can shed light on these intricacies and provide valuable insights moving forward.

## 3. Materials and methods

### 3.1. Participants

A total of 34 university students voluntarily participated in the study with an average age of 22.8 years (SD=±2.5). All of the participants were male. The experiment encompassed two distinct sessions. All 34 participants participated in the first session of the experiment, and 26 in the second session. Data from participants who had participated in one session were also used in the analyses. So, in total, we had 28 participants in the P condition and 28 participants in the NP condition. All participants have normal or corrected to normal vision and hearing. It's worth noting that two participants were excluded due to calibration issues. Participants were eligible for inclusion in the study if they met the following conditions: they were between the ages of 20 and 30, had normal or corrected-to-normal vision, were native Farsi speakers, had no prior familiarity with the experimental materials, were right-handed, and obtained a score of 6–7 on the IELTS Simulator Test (their English listening skills were evaluated using the listening component of the IELTS -International English Language Testing System- simulation test). Participants were excluded from the study if they had prior experience or knowledge of the study material, had recently used alcohol or drugs that could impact cognitive performance, exhibited excessive eye blinking or head movements, showed signs of fatigue or sleep deprivation, or obtained an IELTS Simulator Test score lower than 6 or higher than 7. In adherence to ethical guidelines, all participants provided written consent before the experiment. Notably, the experimental protocols received approval from the ethics committee of the Iran University of Medical Sciences.

## 3.2. Materials

We utilized Lessons 6 and 11 from the Oxford Open Forum 3 [38] textbook to create 4 multimedia versions. The initial two versions, each lasting 290 seconds, were based on Lesson 6. One of these versions adhered comprehensively to five educational multimedia design principles (coherence, signaling, redundancy, spatial contiguity, and temporal contiguity) – referred to as the "P: with principles" condition. Conversely, the other version intentionally deviated from these principles – referred to as the "NP: without principles" condition. Similarly, the 2 subsequent versions spanned 342 seconds and corresponded to Lesson 11. In one version, the same five principles mentioned earlier were dutifully upheld (P). Conversely, the other version disregarded these principles (NP).

## 3.3. Experimental design

The experiment has 6 main stages, the full description of each stage is mentioned in the caption of Fig 1. The participants were divided randomly into two groups of equal numbers (G1 and G2). During the first session, G1 viewed the "P" version of Lesson 6 (P6), while G2 watched the "NP" version of Lesson 6 (NP6). Subsequently, in the second session of the experiment, G1, watched the "NP" version of Lesson 11 (NP11), and G2, viewed the "P" version of Lesson 11 (P11). Before commencing the experiment, all participants received a comprehensive explanation of the experimental procedure. Additionally, they engaged in a simulated experiment to acquaint them with the experimental conditions and methodologies. Once prepared, participants positioned themselves behind the monitor. Before the multimedia's initiation, participants focused on a black circle positioned at the center of a gray screen for 10 seconds. This preliminary step facilitated the collection of eye data about the baseline state. Following this, the designated version from the pool of four multimedia was automatically presented. Afterward, an automatic recall test was administered, wherein participants answered test questions within 420 seconds. Subsequently, participants completed a paper-based NASA-TLX workload questionnaire.

According to the cognitive theory of multimedia learning, five strategies are employed to manage cognitive overload and minimize redundant processing. In our eye-tracking data analysis, we concentrated on three of these strategies within the context of multimedia content. By deliberately violating these principles in the "NP" multimedia version, we aimed to introduce an increased extraneous cognitive load. The three principles are as follows:

1. **Spatial Contiguity Principle:** This principle underscores the importance of arranging text and related graphical content closely to minimize the visual search and the cognitive effort required for scanning back and forth between the text and graphics. In the "NP" multimedia version, we deliberately spatially separated corresponding images and text descriptions within specific frames (see Fig 2-A). This manipulation is anticipated to prompt subjects to continually shift their focus between the graphical and text components as they strive to comprehend the content presented within the scene within a limited timeframe.

2. **Coherence Principle:** This principle advocates eliminating extra content that could distract learners. Only the pertinent information required for mastering the main subject should be included. The coherence principle optimizes cognitive resources by eradicating redundant information [2]. Accordingly, in the "NP" multimedia version, we deliberately introduced additional and unnecessary content in various parts of the scenes within certain frames (refer to Fig 2B). We anticipate that learners' cognitive resources will be distributed among the diverse types of information presented on the screen during multimedia viewing. Consequently, some cognitive resources will be expended to ignore extraneous details [39].

3. **Signaling Principle:** This principle dictates that crucial information should be highlighted to guide learners' attention. By emphasizing significant content, the signaling principle steers learners toward focal points, allowing them to disregard irrelevant details and allocate cognitive resources more effectively to process vital information [2]. For instance, a recent study underscored the enhanced learning outcomes achieved by integrating visual cues in non-procedural

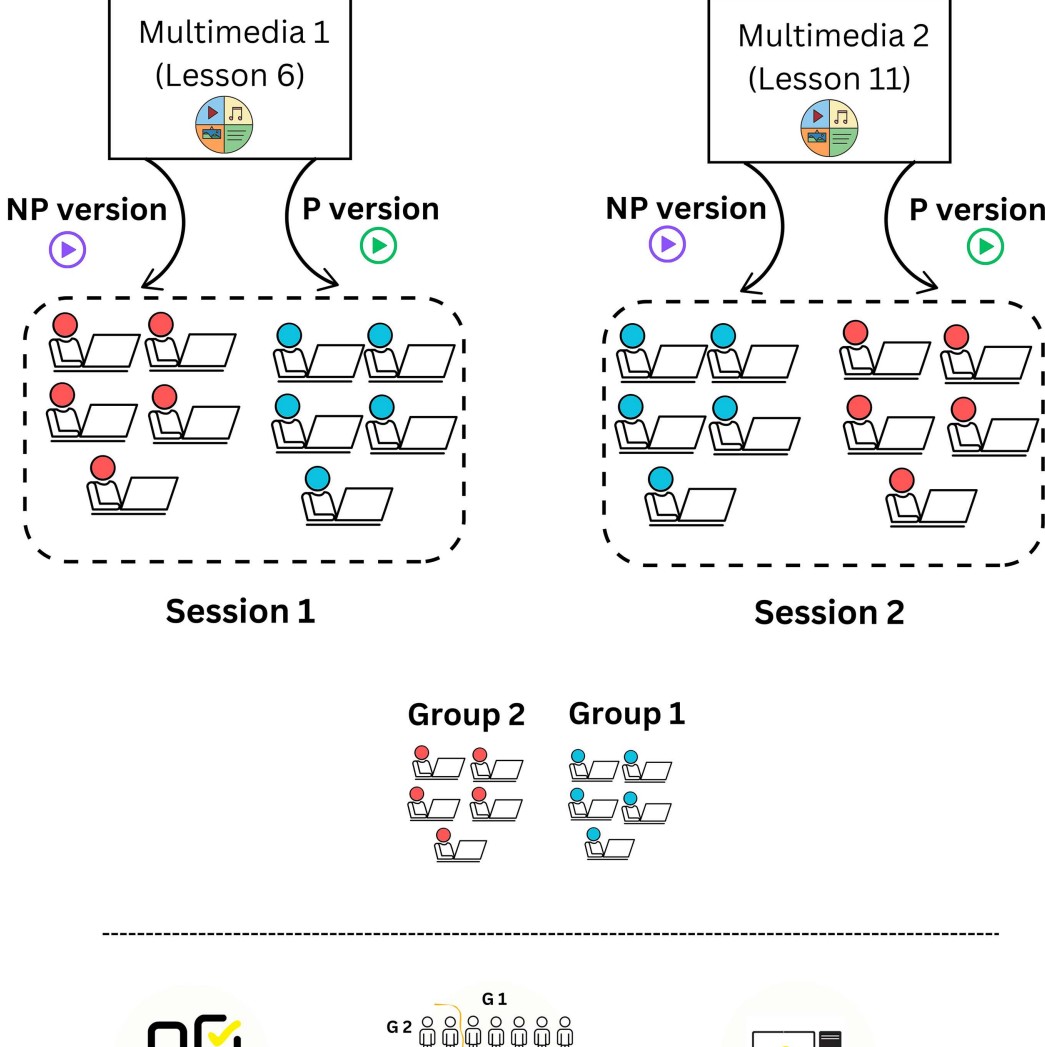

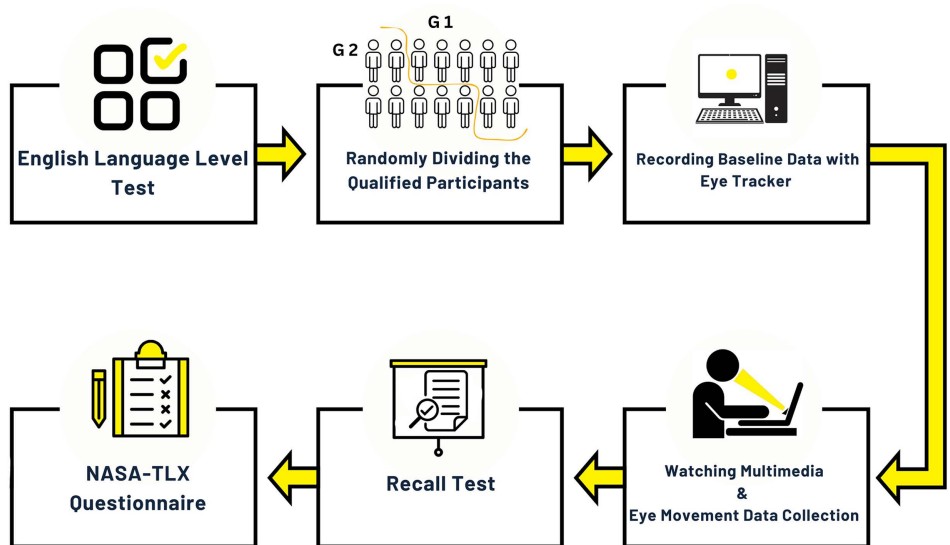

**Fig 1. Experimental design.** The experiment utilized two audio narrations to generate four video variations: P6, NP6, P11, and NP11. For each narration, two versions were produced—one adhering to multimedia design principles (labeled P6 and P11, marked in green) and one violating those principles (labeled NP6 and NP11, marked in purple). Participants were randomly assigned to two groups: Group 1 (G1) viewed the principled version of

the first narration (P6) and the non-principled version of the second (NP11) across two sessions. Group 2 (G2) watched the non-principled version of the first narration (NP6) and the principled version of the second (P11). Experiment steps: First, English language level test (IELTS simulation test); second, randomly dividing the participants who got scores ranging from 6 to 7 into two equal groups; third, looking at a black-filled circle for recording baseline data; fourth, watching the multimedia (no interaction); fifth, taking part in the recall test (via mouse interface); and finally completing the NASA-TLX questionnaire (paper-based version). From the third step, all the steps are completely identical in both sessions. In the third and fourth steps, eye-tracking data are collected. All of these steps were carried out completely consecutively and immediately.

activities within educational videos [40]. Fig. 2C illustrates the signaling principle's application or violation within frames of both the "P" and "NP" multimedia versions.

To ensure the validity of extraneous load manipulation, the NASA-TLX workload index was employed to gauge the workload of participants in both G1 and G2 groups. Renowned for its reliability, this index comprehensively evaluates subjective workload. It consists of six facets: mental pressure, physical pressure, time pressure, performance, effort, and frustration. Given that each facet contributes differently to workload during a specific task, the raw score of each facet is multiplied by its corresponding weight. These weights are determined through 15 pairwise comparisons completed by the participants. The cumulative scores of all six facets are then combined and divided by the number of pairwise comparisons (15) to yield the total workload score [41]. This comprehensive approach guarantees a reliable assessment of the workload experienced by participants in each experimental group.

### 3.4. Eye movement measures

Violating coherence and spatial contiguity principles within multimedia design leads to visual and textual content dispersion throughout the scene. To comprehensively explore the impact of these violations on eye behaviors, we employed the Total Number of Fixations (TNF), Mean Fixation Duration (MFD), and Mean Saccade Amplitude (MSA) measures within specific time of interest (TOI) segments. These measures collectively provide insights into factors like visual effort exertion, the efficacy of visual search, and shifts in additional cognitive load. Below, we provide detailed explanations for each of these metrics:

1. **TNF:** This metric quantifies the total fixations occurring across the screen within a designated TOI [42].

2. **MFD:** MFD represents the average duration of all fixations observed within the TOI across the complete screen [39].

3. **MSA:** MSA denotes the average length of saccades within the TOI across the entire screen [39].

In evaluating concepts such as the findability and noticeability of vital content, as well as the extent of attention on specific screen areas, we calculated the following metrics within AOIs containing crucial learning content:

1. **Time to First Fixation (TFF):** TFF gauges the time between stimulus presentation and the initial gaze directed toward that stimulus. This metric quantifies bottom-up attention originating from stimuli and assesses visual search's effectiveness in evaluating user interfaces [43–44].

2. **Dwell Time (DWT):** DWT measures the cumulative duration participants fixate on a specific AOI. It offers insights into the sustained attention dedicated to particular content within the AOI.

**Fixation Distribution Map** These visual representations showcase the distribution of gaze across various AOIs, providing an intuitive depiction of participants' focus and engagement. This map is constructed by smoothing fixation distributions with Gaussian kernels. It comprises a 3D representation generated from 2D fixation maps, relying on x and y coordinate locations for fixations, while the third dimension represents fixation intensity (weighted by fixation count). This map provides a comprehensive depiction of fixations across the AOIs [42,45].

# Principle

# No Principle

### A: Spatial Contiguity Principle

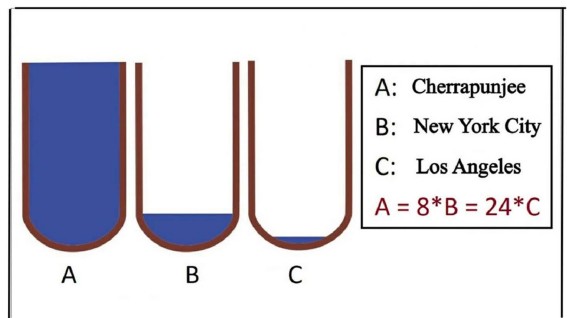

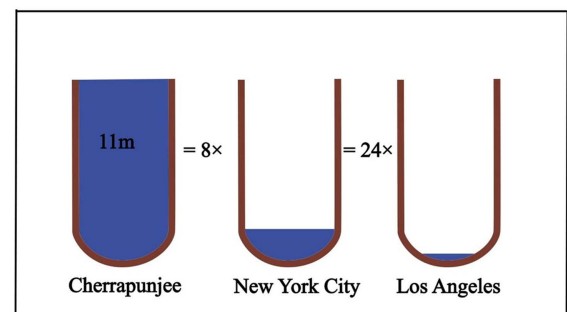

### B: Coherence Principle

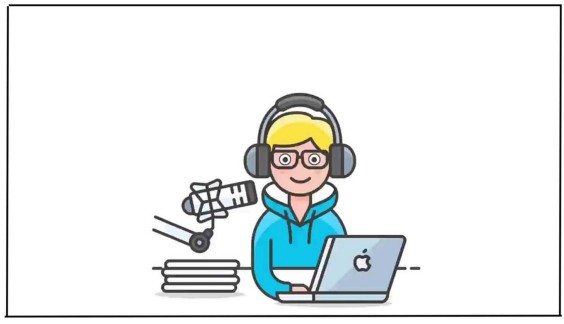

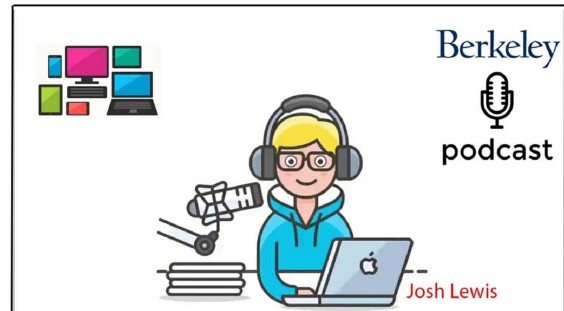

### C: Signaling Principle

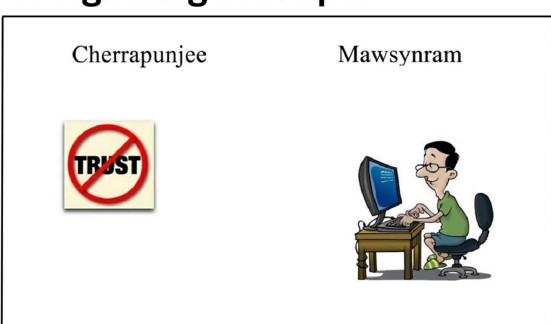

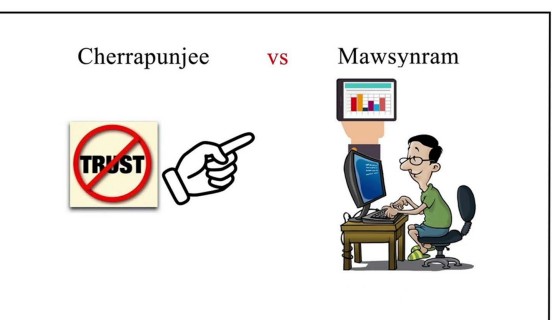

**Fig 2. Educational multimedia.** Three Examples of application (i.e., right panels)/violation (i.e., left panels) of multimedia design principles. Panels A, B, and C are respectively related to the application or violation of the principles of spatial contiguity, coherence, and signaling.

By leveraging these metrics, we can comprehensively understand participants' visual behavior, the attention directed toward significant content, and the cognitive processes accompanying multimedia engagement.

In addition to these metrics, we employed changes in pupil size and microsaccade rate within the designated TOIs corresponding to the observation or violation of the three principles mentioned earlier.

**Pupillary Responses to Cognitive Load** Pupillary responses denote changes in pupil size in response to cognitive demands or mental effort. An increased mental effort typically leads to pupil dilation. Pupillary responses are utilized in research to assess mental workload and attention and are sensitive to diverse cognitive tasks and mental states [46–47]. Microsaccades Responses to Cognitive Load: Microsaccades, akin to pupillary dilation, can be investigated from an information processing perspective. Numerous studies indicate an inverse relationship between microsaccade rate and cognitive load. Siegenthaler and colleagues, for example, established that heightened task complexity correlates with a reduction in microsaccade rate [24].

These comprehensive metrics collectively facilitate a deeper understanding of participants' cognitive processes, attention allocation, and engagement dynamics during multimedia interactions.

### 3.5. Recall test

Given the likelihood of learners' performance being influenced by the design of the educational video they watched, we employed a recall test after multimedia viewing to gauge and compare the learning outcomes between the "P" and "NP" conditions.

Upon completion of the multimedia engagement, the software integrated into the laboratory computers automatically initiated a recall test. This test comprised 12 multiple-choice questions about the content covered within the multimedia. The questions were meticulously crafted to encompass crucial points from the material, facilitating a comprehensive evaluation of participants' grasp of the subject matter. Respondents were required to provide answers using a mouse, navigating between questions while being restricted to viewing one question at a time. A time frame of 420 seconds was allotted for answering the questions. Participants could leave questions unanswered or modify their responses during this duration. This recall test methodology assessed participants' comprehension and learning progress.

### 3.6. Data analysis

We employed a video-based monocular eye tracker, specifically the Eyelink-1000 Plus model, to track eye position and pupil size. Stimuli were presented on a 17-inch LCD panel with a 1280 × 1024 pixels resolution. The sampling frequency of the eye tracker was set to 1000 Hz and data were obtained from both eyes of the participants. The experiment was conducted in a dimly lit room with no direct light sources. We used a standard 9-point calibration. It required participants to fixate on a series of known points, allowing us to map eye positions to specific locations.

In line with established practice, fixations of less than 90 ms were deemed too brief for meaningful information processing and, thus, were excluded from the analysis. because they were more likely to be artifacts, noise, or involuntary eye movements. Following existing research methodologies [48–51], we employed linear interpolation scipy's interpolate module to replace pupil values during blink episodes in which we had some missing data. Furthermore, the raw pupil signals underwent smoothing using a low-pass digital filter to enhance signal quality. We used Savitzky-Golay filter, that fits a polynomial to a local window of data using least square regression. We set the degree of the polynomial to 3 which provides better results in rapid changes while still avoiding overfitting. Regarding the pupillary response to changes in luminance, it's noteworthy that only a single fixed frame was presented within all the selected TOIs. Moreover, the brightness of these frames remained consistent between the P and NP conditions. This brightness uniformity was supported by the outcome of two-tailed paired t-tests conducted on the brightness of the frames. The result of this test demonstrated no significant difference ($t = 0.243$, $p = 0.83$) between the two conditions' frame brightness.

It's also crucial to acknowledge that the frames preceding the TOIs' onset were identical across both the P and NP conditions. Consequently, considering the uniformity of the frames and the lack of significant difference in brightness, the expectation was that the variations in pupil size attributable to lighting conditions between the two conditions would not exhibit substantial differences. For the detection of microsaccades, we adapted Engbert and Kliegl's algorithm [52]. Our choice of parameters included a velocity threshold of $\lambda = 5$ and a minimum duration threshold set at 6 ms. To evaluate pupillary responses and microsaccade rates, we executed paired t-tests to test our hypothesis that manipulating cognitive load and allocating attentional resources trigger pupillary and microsaccadic responses. We conducted an FDR (False Discovery Rate) correction to mitigate the possibility of false discoveries, ensuring a controlled probability of such errors. By adhering to these rigorous methodological steps, we ensured a comprehensive and reliable analysis of pupillary responses and microsaccade rates in response to changes in cognitive load and attentional allocation strategies.

## 4. Results

This section comprehensively analyzes participants' performance, subjective cognitive load estimation, and collected gaze data from two experimental sessions.

### 4.1. NASA-TLX and recall test

To assess the differences between the P and NP conditions, we conducted an independent samples t-test on the students' NASA scores and recall performance. To assess the normality of the data distribution visually, Q-Q (Quantile-Quantile) plots were generated. The points closely followed the reference line, suggesting no substantial deviation from normality. For the NASA-TLX scores, skewness (0.45) and kurtosis (0.37) fell within the acceptable range (−1 to +1), while for the Recall scores, skewness (−0.47) and kurtosis (0.6) were also within this range. These results indicate approximate symmetry and a mesokurtic distribution. The normality of the data was also assessed using the Kolmogorov-Smirnov test. The results indicated that the data were consistent with a normal distribution (the tests have more than 0.05 p-value suggesting that the data are consistent with normality). Levene's test for homogeneity of variance indicated that the variances were equal across groups, suggesting that the assumption of homogeneity of variance was met. We anticipated observing meaningful distinctions between these two conditions. Illustrated in Fig 3, the participants in the NP condition indicated a notably higher workload than those in the P condition (t(54) = −6.223, p = 1.06E-08). Furthermore, the P condition

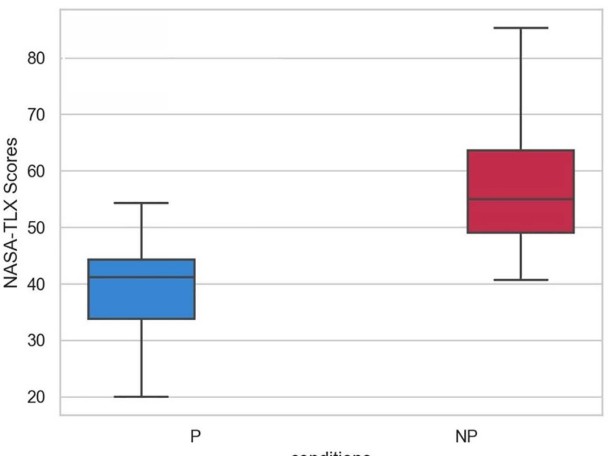 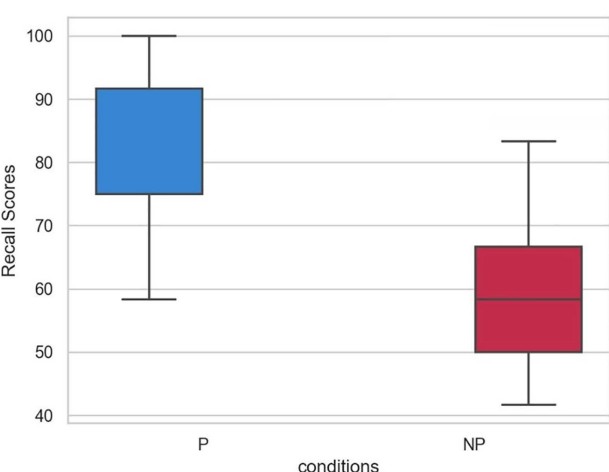

**Fig 3. Result of NASA-TLX and Recall test.** The scores are in the range of 0 to 100. Middle line represents the median (50th percentile) of the data. Whiskers (error bars) extend to the minimum and maximum values within 1.5 × IQR from Q1 and Q3.

performed significantly better on the recall test than the NP condition (t(54) = 5.325, p = 2E-06). To compare the two conditions, the non-parametric Mann–Whitney U test was also applied. For the NASA scores, the test yielded U = 74, p < 0.001, indicating a statistically significant difference between the conditions. Similarly, for the Recall scores, U = 674.5, p < 0.001, also indicating a significant difference.

### 4.2. Regression analysis

In the initial eye movement data analysis, we identified the TOIs which were pertained to observing or violating coherence and spatial contiguity principles. Subsequently, for each participant, we quantified three metrics: TNF, MFD, and MSA within each TOI.

To delve deeper, we employed linear regression analysis to explore potential connections between these metrics and participants' performance in the test and their perceived cognitive load. In order to assess the assumption of normality for the residuals in our linear regression model, we conducted the Shapiro-Wilk test. The results showed that the residuals did not significantly deviate from a normal distribution (all p-values > 0.05). Thus, we conclude that the residuals of the model are approximately normally distributed. The complete results of regression analysis are comprehensively presented in Table 1 and Table 2.

Table 1 illustrates a significant correlation between NASA-TLX scores and all three measures: TNF, MFD, and MSA. This correlation suggests that as the number of fixations increases and saccade sizes become larger, while fixation durations decrease within TOIs, participants experience heightened cognitive and mental load on their working memory. This underscores the sensitivity of the cognitive load perceived by individuals to all three metrics.

Furthermore, as indicated in Table 2, learners with lower TNF and MSA measures achieved notably higher scores on the performance test. Conversely, learners with higher MFD measures achieved significantly lower scores. This implies that an increased number of fixations within limited TOI and extensive movement across distant screen areas can have a relationship with decreasing fixation duration. This, in turn, can impact on overall cognitive load, resulting in poorer performance on the test.

**Table 1. Summary Table of linear regression analysis results for predicting NASA-TLX.**

| Eye Movement Measures | NASA – TLX | R² |
|---|---|---|
| TNF[a] | 5.275** | 0.162 |
| MFD[b] | −4.658** | 0.133 |
| MSA[c] | 7.617*** | 0.356 |

[a]constant: 37.450

[b]constant: 49.281

[c]constant: 49.281

**Table 2. Summary table of linear regression analysis results for predicting recall scores.**

| Eye Movement Measures | Recall Scores | R² |
|---|---|---|
| TNF[a] | −8.294*** | 0.246 |
| MFD[b] | 8.655*** | 0.283 |
| MSA[c] | −8.315*** | 0.261 |

[a]constant: 91.519

[b]constant: 72.916

[c]constant: 72.916

Our subsequent analysis involves eye movement data from TOIs associated with the observation or violation of the signaling principle. In these TOIs, essential graphical content for learning is highlighted through techniques such as motion, color changes, and graphical cues to attract users' attention. We identified AOIs within each frame to pinpoint these critical components. We measured TFF and DWT for each TOI. We performed a logistic regression analysis to understand the relationship between these measures and recall scores.

Concerning the content within each TOI, a corresponding question was included in the recall test for Lesson 6. Correct responses to these questions were assigned a score of one, while incorrect answers received a score of zero. The outcomes of this analysis are documented in Table 3. The regression model employed as follows:

$$Test\ score = B_0 + B_1 \times DWT + B_2 \times TFF \tag{1}$$

Based on the outcomes of this analysis, where the pseudo-R2 value is 0.072, noteworthy relationships have been identified. A significant positive correlation emerged between learning test scores and DWT, while a significant negative correlation was observed between learning test scores and TFF. This suggests that students who quickly focus on AOIs and allocate more time to observe them tend to perform better in the final learning test.

Continuing our exploration, we undertook a linear regression analysis to delve into the connections between DWT and TFF regarding compliance/violation of the signaling principle. The results of the regression analysis unveiled substantial associations. Specifically, a positive and significant relationship was established between DWT and the Observing signaling principle ($F(1,166) = 27.99$, $R2 = 0.144$, $B = 0.185$, $p < 0.001$). Conversely, TFF exhibited a negative relationship with the Observing signaling principle ($F(1,166) = 165.4$, $R2 = 0.499$, $B = -0.443$, $p < 0.001$).

These findings highlight the predictive nature of observing or violating the signaling principle about DWT and TFF among students. Moreover, these results provide compelling evidence of the significant influence of the "cuing effect" on learning within educational content.

## 4.3 Microsaccadic rate

Initially, we validated the accuracy of the algorithm in detecting microsaccades. To accomplish this, we executed a correlation analysis involving peak velocity and amplitude, a measure referred to as the "main sequence." This analysis is grounded in the established understanding that microsaccades exhibit a positive correlation between these two parameters, like regular saccades.

Our investigation yielded a notably strong positive correlation ($r = 0.913$, $p < 0.001$), affirming the correct identification of microsaccades. This finding provides confidence in the algorithm's ability to detect microsaccades, as depicted in Fig 4, accurately.

We investigated the impact of compliance/violation of principles on microsaccade responses. We calculated the microsaccadic rate within a 100-ms temporal interval, commencing from initiating the TOIs. As part of this analysis, we established a baseline rate by averaging the rates within the epoch spanning −100 to +100 ms relative to the onset of the TOIs. This involved the individual computation of microsaccadic rates for each participant, followed by the averaging of results across all participants.

**Table 3. Summary table of logistic regression analysis results for predicting recall scores.**

| Eye Movement Measures | Recall Scores |
|---|---|
| DWT | 1.668* |
| TFF | −1.565** |

[a]intercept: −1.307

[b]* P < .05 ** P < .01 *** P < .001

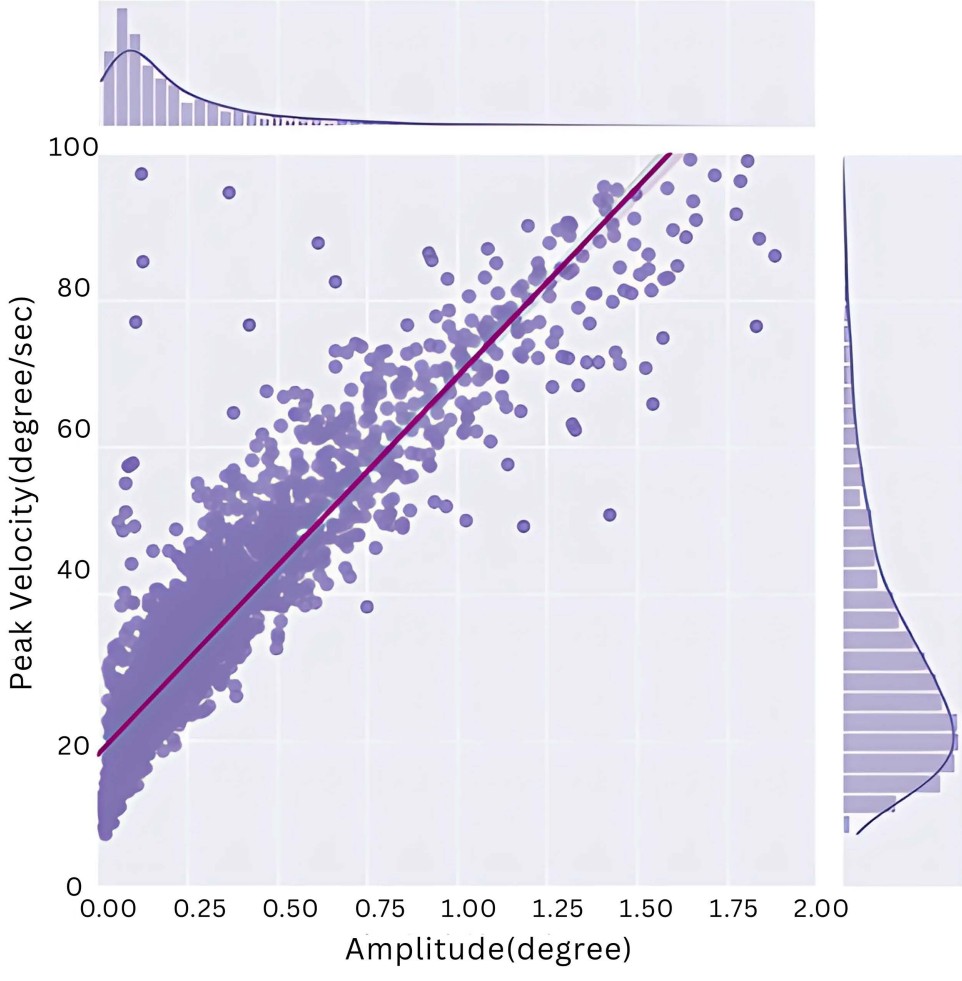

**Fig 4. Microsaccadic peak velocity vs. magnitude (main sequence).** This correlation established that microsaccades were correctly detected.

For a more comprehensive understanding of microsaccade behavior, we conducted multiple comparisons, corrected for false discovery rate (FDR), between the mean microsaccadic rates observed in the P and NP conditions. These comparisons were performed within a 300-ms moving window, initiated at the onset of the TOIs. This methodology aligns with similar approaches documented in references [53–54].

The results revealed that the rate of microsaccades during TOIs was notably lower in the NP condition compared to the P condition. This observation underscores distinct differences in microsaccade patterns between the two conditions in response to the manipulation of principles. As depicted in Fig 5, specifically in panel A, comparing baseline and initial microsaccade rates following the commencement of the TOIs revealed no significant differences between the two conditions. However, as we progressed within this time interval, the rate of microsaccades experienced a decline. It decreased from approximately 2.2 to around 1 within the NP condition. Notably, throughout the TOIs, the microsaccade rate in the NP condition consistently remained significantly lower than in the P condition. This pattern persisted until the conclusion of the TOIs, marked by the gray vertical line. It's worth noting that all FDR-corrected comparisons indicated significant differences between the two conditions ($ps < 0.02$ and $ts > 7.36$), except for the first two 300-ms time windows (0–300 ms

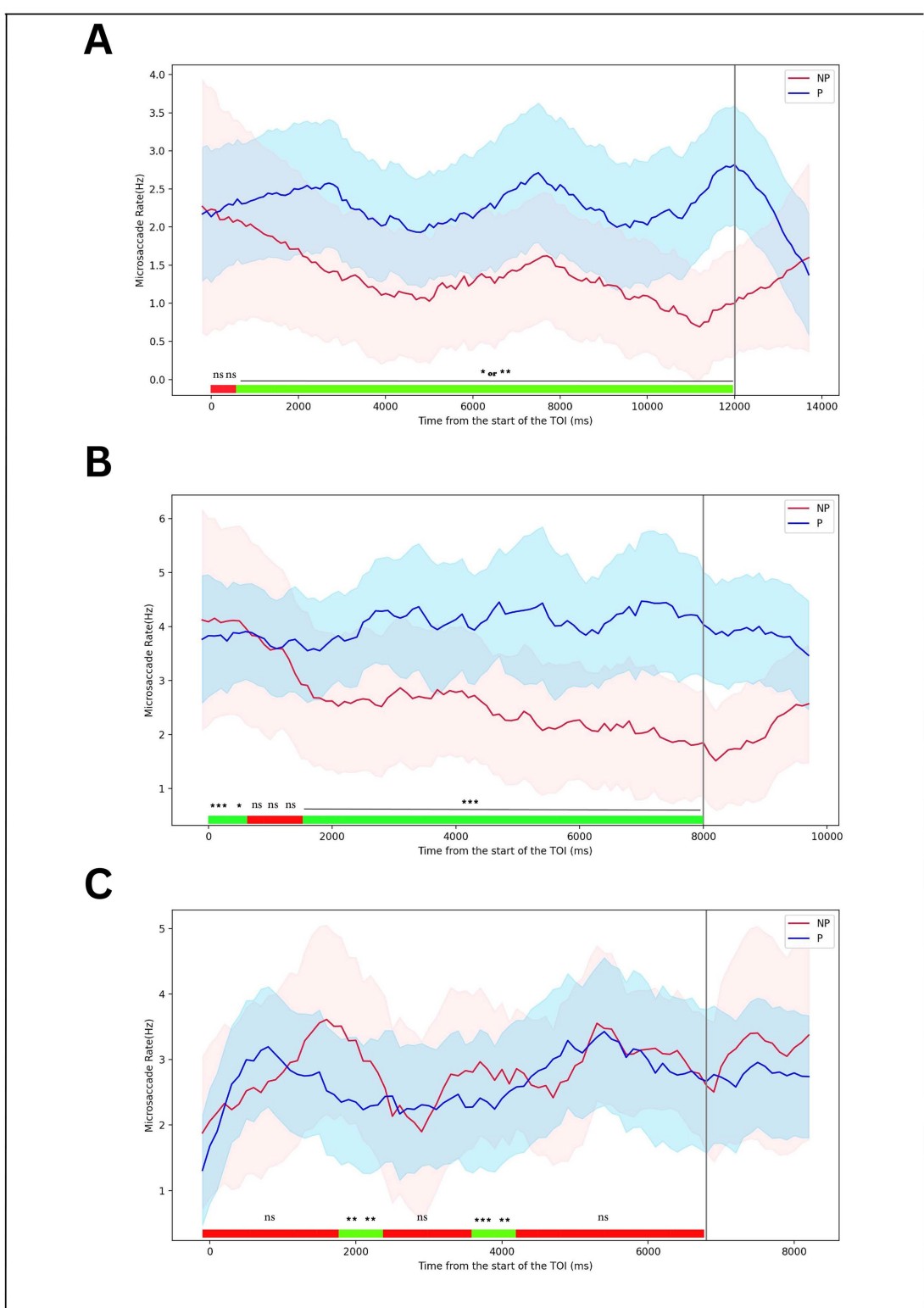

**Fig 5. Microsaccadic rate calculated within the TOIs.** Shaded areas indicate the standard error of the mean. Green and red rectangles above the *x*-axis indicate the 300-ms time windows (FDR-corrected) used for statistical testing. Asterisk denotes a significant difference between the two experimental conditions, while "ns" means the difference was non-significant.

and 300–600 ms), where the comparisons showed non-significant results (ps > 0.18 and ts < 1.41), as represented by the red bars beneath the graphs.

Moving to panel B, during the baseline state, the microsaccade rate of the NP condition was initially higher than that of the P condition. However, after the initiation of the TOIs, the microsaccade rate of the NP condition experienced a decline, evident around the 800-ms mark. At the onset of this suppression, the microsaccade rates in both condition were equal. Subsequently, the microsaccade rate of the NP condition continued to decrease. Consequently, the frequency of microsaccades in the NP condition gradually diminished, sustaining this gradual decline until the conclusion of the TOIs.

In terms of statistical significance, within the NP condition and P condition comparisons, there were three 300-ms time windows (600–900 ms, 900–1200 ms, and 1200–1500 ms) where the FDR-corrected comparisons did not demonstrate significant differences (indicated by the red bars in Fig 5B; ts < 2.10, ps > 0.17). Conversely, outside these time windows, all comparisons reached canonical levels of significance (indicated by the green bars in Fig 5B; ts > 15.92, ps < 0.02). In general, within TOIs where the principles of multimedia design are not adhered to, there is a noteworthy reduction in the microsaccade rate compared to the condition following the principles. This decrease is observed concerning the baseline level. Notably, the TOIs depicted in Fig 5A and Fig 5B pertain to coherence and spatial contiguity principles. Conversely, no significant difference is detected in the microsaccade rate between the conditions when considering TOIs associated with the signaling principle.

Fig 5C is dedicated to evaluating compliance or violation of the signaling principle in one of the testing and observation conditions (TOIs). As the illustration reveals, there is no discernible pattern in the fluctuations of microsaccade rates within the two conditions. Additionally, no statistically significant distinction is observable between these rates.

## 4.4. Pupil size

An increase in pupil size could be a sensitive indicator of mental effort and cognitive load. To test this hypothesis, the following approach was employed:

1. **Baseline Calculation:** A baseline value was established by averaging the pupil size data from −200 ms to +100 ms relative to initiating the TOIs. This baseline value was used as a reference point for subsequent analysis.

2. **Baseline Correction:** Each data point was subtracted from the corresponding baseline value, following the application of the baseline correction procedure. This procedure was executed independently for each participant regarding individual differences, allowing for accurate comparison.

Continuing the analysis, pupil size within the TOIs underwent a similar procedure as that of the microsaccade rate. Specifically, ten FDR-corrected comparisons were conducted between the mean pupil size changes for the P and NP conditions. These comparisons were carried out using a 300-ms moving window, commencing at the onset of the TOIs.

In Fig 6A, the pupil size changes observed in the NP condition exhibited a distinct pattern. Roughly 1500 ms following the initiation of the TOI, a pronounced expansion in pupil size peaked at around 2000 ms, representing a notable increase of 300 units compared to the baseline pupil size. Following this expansion, there was a subsequent drop, leading to a suppression of pupil size until approximately 3800 ms. This suppression eventually reached a relatively stable state. In contrast, the pupil size changes in the P condition did not demonstrate significant variations compared to the NP condition. Across all 300-ms windows, the results of FDR correction consistently demonstrated significant differences between the two conditions. When the pupil size of the NP condition was smaller than that of the P condition, p-values were less than 0.001, and t-scores were greater than 71.43. Conversely, when the pupil size of the NP condition was larger than that of the P condition, p-values were less than 0.001, and t-scores were less than −10.

Both conditions' graphs in Fig 6B displayed a rapid expansion following a minor contraction. This expansion reached its zenith approximately 1000 ms after the commencement of the TOI. Subsequently, there was a gradual decrease, around 2500−3000 ms, after the TOI began, leading to an approximately stabilized state. Notably, the size of changes in the

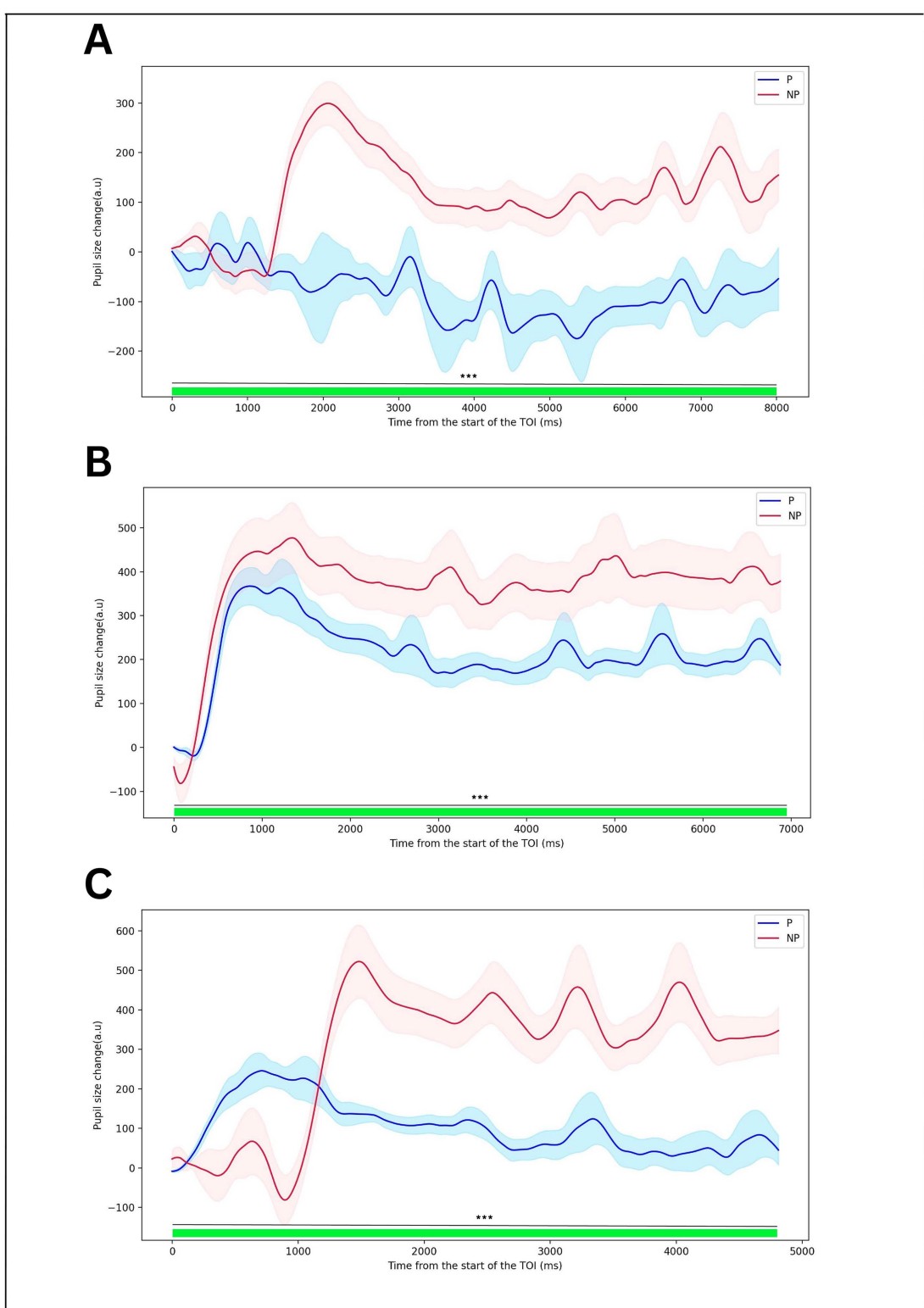

**Fig 6. Change in pupil size calculated within the TOIs.** Shaded areas indicate the standard error of the mean. Green and red rectangles above the *x*-axis indicate the 300-ms time windows (FDR-corrected) used for statistical testing. Asterisk denotes a significant difference between the two experimental conditions, while "ns" means the difference was non-significant.

NP condition was more substantial, featuring a more pronounced expansion than the P condition. Throughout the TOI, significant differences between the two conditions were consistently observed. Specifically, within the first 300-ms window, FDR correction results indicated p-values below 0.001 and t-scores exceeding 8.80. Beyond the initial window, for the remaining time windows, p-values were below 0.001, and t-scores were below −83.06. In panel C of the analysis, distinct patterns in pupil size changes were observed. In the NP condition, a momentary suppression was noticeable, succeeded by a rebound that occurred around 950 ms and peaked approximately 1500 ms after the onset of the TOI. Conversely, in the P condition, the pupil size changes increased from the start of the TOI, reaching its peak at roughly 1000 milliseconds. Subsequently, there was a gradual decline, leading to a stable state at around 2000 milliseconds.

The FDR correction results for various time windows revealed significant differences between the NP and P conditions. During the time windows when pupil size changes were smaller in the NP condition than in the P condition, p-values were below 0.001, and t-scores exceeded 9.20. For time windows where pupil size changes were larger in the NP condition compared to the P condition, p-values remained below 0.001, and t-scores fell below −55.23.

These observations collectively indicate that larger changes in pupil size were consistently observed in the NP condition compared to the P condition across all TOIs associated with the three principles of contiguity, coherence, and signaling.

### 4.5. Fixation distribution map

We analyzed fixation distributions in signaling TOIs for all participants, creating fixation distribution maps. The outcomes are illustrated in Fig 7, where fixation distribution maps for both participant conditions are displayed across panels A, B, C, and D. To illuminate disparities in attention allocation, we subtracted the gaze distribution maps of the P condition from those of the NP condition in the third column of the first row. The second row showcases frames associated with TOIs, denoted by black boxes marking AOIs. Notably, certain AOIs linked to the signaling principle in this study exhibit dynamic attributes, such as motion or color changes, which may need to be discerned in static images.

In Fig 7A, during the discussion of India's economic growth, the arrow in the image gradually transitions to red within the P condition's multimedia. Evident from the 2D density map of condition P, this multimedia element effectively captures learners' attention. Fig 7B involves the explanation of the geographical locations of Indian states. Within the P condition's multimedia, this concept gains emphasis by placing a compass on the map, which intermittently flashes toward the northwest direction. In contrast, the NP condition's multimedia solely displays the country's map.

Fig 7C entails the speaker's account of their journey to the subcontinent, wherein the P condition's multimedia highlights this episode through a red marking, accompanied by the appearance of the word "subcontinent" within the image. Fig 7D discusses local winds known as monsoons. In condition P, the key content's prominence is achieved through the dynamic movement of curves representing the winds, accompanied by displaying the wind's name adjacent to these curves.

As anticipated, across all the TOIs examined in this analysis, the count of fixations within the designated AOI was notably higher for the P condition. In this condition, the main content was specifically highlighted to capture learners' attention. The P condition consistently demonstrated this superiority in focused attention on crucial screen areas compared to the NP condition. Conversely, participants in the NP condition exhibited more scattered fixations, likely due to their inability to identify the primary content.

### 4.6. Visualizing gaze behavior

To gain insights into players' visual attention and interaction patterns during watching multimedia, we analyzed eye-tracking data using scan paths and fixation distribution map. Scan paths illustrate the sequential movement of gaze, highlighting the order and transitions between fixations, in Fig 8, you can see an example of a scan path of a TOI in which the spatial contiguity principle observed/violated.

 

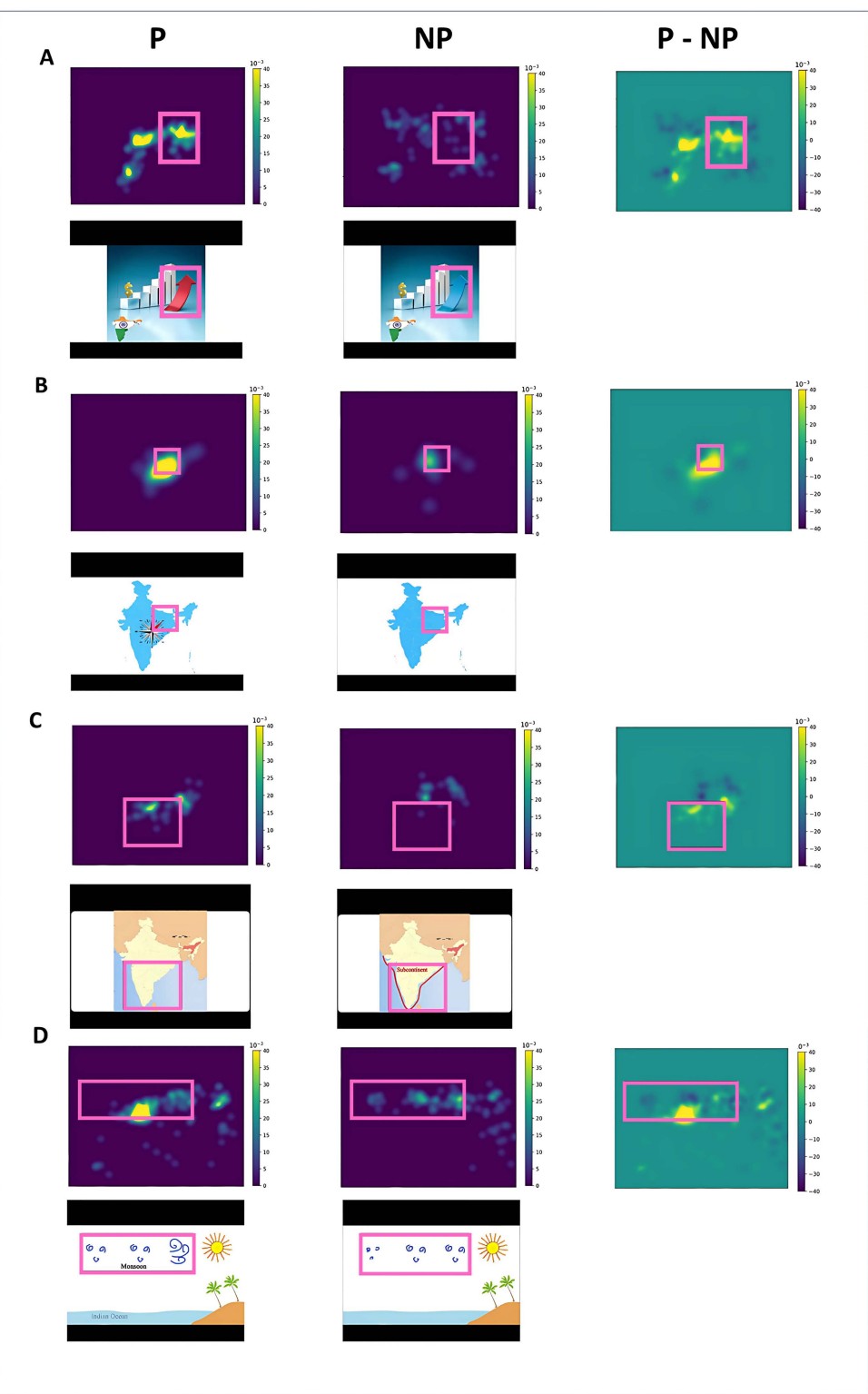

**Fig 7. Fixation Distribution Maps.** Axes values represent number of fixations. Each panel shows a TOI. The first and second images in each panel are related to the fixation distributions of the two P and NP conditions. Higher fixation probability is represented by yellow. The third picture shows the difference probability maps of the P-NP, with yellow (positive values) indicating a higher fixation probability for P than NP. The fourth and fifth pictures (two pictures of the second row in each panel) show Multimedia frames presented in TOIs for P and NP conditions.

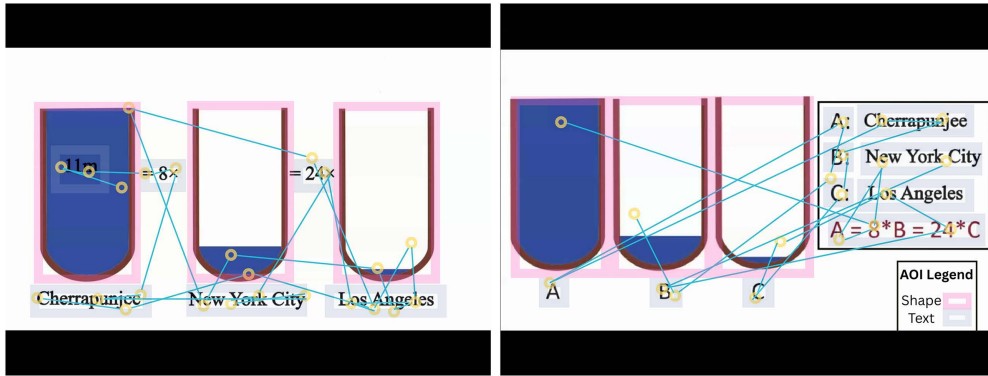

**Fig 8. Scan the path of two subjects while watching multimedia with (left) and without (right) the application of the Spatial Contiguity Principle.**

Fixation distribution maps provide an aggregated view of fixation density across different regions of interest. These visualizations allow us to identify key areas that captured user's attention, common gaze patterns, and variations in attentional focus based on dynamics of Compliance and non-compliance with multimedia design principles. In Fig 9 we present the result of fixation distribution map analyses, demonstrating how users visually engage with the multimedia.

Cronbach's alpha was computed to assess the internal consistency of the eye movement metrics utilized in this study. Given the expected variability in eye movements between the P and NP conditions, separate analyses were performed for each. As shown in Table 4, the results indicate a high reliability level across most tested eye movement metrics.

## 5. Discussion

The primary objective of this study was to examine the impact of adherence to or divergence from Mayer's multimedia design principles on learning effectiveness, cognitive load experienced by learners, their eye behavior, and, subsequently, the allocation of attention to various presented content. This goal was accomplished by analyzing participants' eye-tracking data, performance test outcomes, and responses to the NASA workload questionnaire.

The NASA-TLX questionnaire was initially employed to gauge subjective cognitive load during multimedia and the findings indicate that non-compliance with multimedia design principles significantly heightened the perceived cognitive load among individuals.

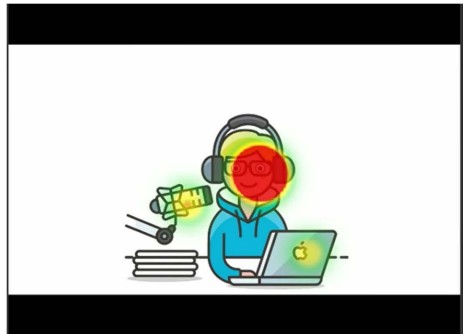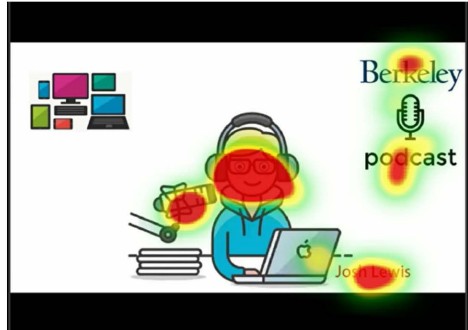

**Fig 9. Attention distribution of two subjects while watching multimedia with (left) and without (right) the application of the Coherence Principle.**

**Table 4. Cronbach's Alpha values for the eye movement measures.**

| Eye Movement Measures | Cronbach's alpha | |
|---|---|---|
| | P | NP |
| TNF | 0.750 | 0.890 |
| MFD | 0.711 | 0.620 |
| MSA | 0.710 | 0.618 |
| TFF | 0.844 | 0.813 |
| DWT | 0.266 | 0.201 |

Likewise, in examining recall test results, we assert that the educational content's design influences student performance on the recall test. Upon evaluating the test scores, a distinct pattern emerged and the results underscore the robust connection between recall test scores and subjective cognitive load as measured by the questionnaire; both were closely tied to adherence or non-adherence to multimedia design principles. Subsequently, we employed linear regression analysis to assess the relationship between eye movement metrics and Recall and NASA test scores.

While prior studies [19,55–58] have proposed that fixation-related metrics can indicate mental workload, attention, and search efficiency, divergent outcomes have been observed in different research findings. Nevertheless, the sensitivity of fixation-related metrics to cognitive load hinges upon factors such as design methodology and content complexity [59] As noted by Jacob and Karen [42,60], the greater the number of fixations becomes, the less the viewer's search for information on the computer screen is assumed to be.

The elevated count of fixations observed within the NP condition can signify diminished target findability or reduced target accessibility, attributed to numerous unrelated distractor elements where coherence principles are violated [61–62]. The elevated count of fixations also arises from frequently shifting gaze between disparate visual and textual information sources. A depiction of this can be found in Fig 8, representing the scan paths of two subjects from both P and NP conditions concerning the spatial contiguity principle.

Findings from various studies indicate divergent patterns in how fixation durations change under conditions of high cognitive load. Some studies have shown that increased cognitive load leads to longer fixation durations as individuals spend more time processing information [63]. This is typically observed when individuals are faced with complex or unfamiliar stimuli that require additional cognitive resources for understanding or decision-making. In such scenarios, the brain's need for greater information processing may manifest as longer fixations, particularly on critical aspects of the task.

On the other hand, other research suggests that higher cognitive load can lead to shorter fixation durations in some contexts [57,64]. This is often seen in tasks where individuals are overwhelmed by excessive information or are under time pressure. In such cases, participants might demonstrate a more superficial level of processing with shorter, more frequent fixations, attempting to process information rapidly without delving deeply into any one element.

Several factors can contribute to these varying patterns of fixation duration under high cognitive load like:

Task Complexity: More complex tasks that require deep cognitive processing tend to lead to longer fixations, while tasks with simpler or familiar components might result in shorter fixations.

Information Overload: When cognitive resources are taxed by excessive or overwhelming information, the individual may reduce the duration of each fixation to quickly scan and process multiple elements.

Our experiment's dynamic, video-format educational content imposed temporal constraints on frame and content display. Consequently, the anticipated increase in fixations within the NP condition was expected to lead to shorter fixation durations, aligning with the outcomes observed. The reduction in fixation duration in the NP condition should not be misconstrued as an indication of content simplicity or ease of processing. Rather, it points toward shallower and more superficial processing [39].

Hyona [65] proposes one approach to assess users' visual search efficiency and ability to identify crucial on-screen information: analyzing the dispersion of fixations. In the NP condition, the fixations were widely dispersed and distanced from each other due to the diffusion of distinct content elements across the screen. The elongated saccades in the NP condition stemmed from the spatial dispersion of various content components, as illustrated in Fig 9, depicting the Attention distribution heatmap of two subjects during the multimedia presentation. The violation of the spatial contiguity principle led to a separation between graphical and textual content, confounding learners and necessitating transitions between different elements.

It's noteworthy that, contrary to general trends, an increase in cognitive load results in shorter saccade lengths [66–67] and increased average fixation durations [68]. However, as explained earlier, our counterintuitive results within the NP condition don't signify simplified processing, reduced mental load, or efficient visual searches. Drawing upon references [8,69], two pivotal strategies for reducing extraneous cognitive processing, thereby augmenting resources for essential and generative processing, are the spatial contiguity principle and the coherence principle. It is posited that when content is presented disparately, greater visual searching becomes imperative, necessitating cognitive resources for upholding individual elements within working memory before their mental integration. This influx of extraneous processing consumes resources that would otherwise be available for crucial and generative processing.

Furthermore, incorporating seductive details within multimedia can divert students' attention, complicating the organization and integration of vital learning content. Consequently, introducing extraneous material amplifies extraneous processing, depleting cognitive working memory resources indispensable for essential and generative processing.

Consequently, the violation of these multimedia design principles subjects users to many challenges, including working memory overload, disregarding seductive details and managing visual search, and the endeavor to sustain focus on primary content while processing information. We also employed logistic regression analysis to explore the influence of signaling principles on individuals' learning and performance. As elucidated in the literature [42], TFF metric enables the assessment of AOI noteworthiness and findability, indicating how swiftly an AOI captures attention.

Given that AOIs were observed faster in the P condition's multimedia than in the NP condition, we can infer that adherence to the signaling principle rendered vital areas conspicuous. These zones' accentuation and prompt detection translated into enhanced test performance. These outcomes resonate with findings from earlier investigations [8,70–72].

According to the result obtained for DWT metric users are drawn towards AOIs through the signaling principle, comprehending that these areas furnish significant information. The prolonged time dedicated to AOIs arises from the processing time allocated to these pivotal regions, ultimately culminating in superior performance in the learning test. These findings align with previous research [8,70–72].

This utilization of logistic regression analysis enables a deeper understanding of the impact of signaling principles on individuals' learning and performance, affirming the significance of these principles in shaping effective multimedia design.

To explore the connection between mental effort and cognitive load with the count of microsaccades and pupil size variations compared to baseline levels, we assessed these parameters within TOIs related to spatial contiguity, coherence, and signaling principles. The findings are summarized below:

**Microsaccade Rate:** In TOIs corresponding to spatial contiguity and coherence principles, the microsaccade rate exhibited a gradual decrease within the NP condition. Prior studies have demonstrated that task load exerts influence over the microsaccade rate. Specifically, increased visual load correlates with an elevated microsaccade rate, while augmented mental load is associated with a diminished microsaccade rate [26,53,73]. Also, in the TOIs related to violating the signaling principles for the NP condition, we do not observe significant changes. Consequently, further research is needed to validate these findings based on the microsaccade rate in the context of multimedia learning and complex video-based representations.

**Changes in pupil size:** Across all analyzed TOIs, a significant dilation of pupil size was observed in the NP condition, occurring approximately after ~1000 ms from initiating the TOIs. It is worth noting that, as we mentioned before, the

frames preceding the TOI start are the same in both conditions. Additionally, the frames associated with TOI demonstrated no significant disparity between the two conditions concerning brightness levels. Consequently, the pronounced difference in pupil size alterations between the two conditions stemmed from an alternate factor, namely cognitive load and the participants' mental engagement. These outcomes align seamlessly with prior research findings [74–77].

As indicated by the results concerning the 2D density map, our expectations were fulfilled. Using the signaling principle successfully directed learners' attention to vital content within all TOIs. The elevated count of fixations within the AOI within the P condition, relative to the NP condition, underscored the conspicuousness and ease of locating the designated AOI in the P multimedia. Consequently, it reduced the necessity for extended visual searching to identify them.

In a broader context, it is plausible to argue that under the premise of dual coding and the Cognitive Theory of Multimedia Learning (CTML), applying the signaling principle helps students eliminate extraneous content. Simultaneously, the signaling principle adeptly organizes pertinent graphical and textual information, diminishing the need for exhaustive scene exploration and enhancing focus on principal themes. This strategic approach optimizes working memory capacity and significantly alleviates cognitive load during multimedia consumption.

## 6. Conclusion

In this study, we investigated the effectiveness of multimedia educational content design principles using eye behavior data as a physiological measure, the NASA-TLX test as a self-reporting measure, and the recall test as a performance measure. The possibility of evaluating the cognitive load imposed on people's working memory and the amount of visual attention was investigated using different analyses. First, in order to investigate the effect of violation/observance of the design principles, we extracted various features from the eye movement data to measure various concepts such as visual attention, cognitive load, findability, and visual search strategy of users in the scene. Then, by applying linear and logistic regression models, we investigated the relationship between these eye-tracking measures and NASA-TLX scores and recall tests. This study also delves into microsaccade rates and changes in pupil size, dissecting learners' eye behavior in response to compliance with or violating the principles. Each of these analyses was performed separately within distinct TOIs. We analyzed fixation distributions to show the concentration of fixations in certain areas of the scene. These analyses can significantly help educational multimedia designers in creating multimedia by imposing the optimal amount of cognitive load on learners' working memory and drawing more attention to critical content.

## 7. Limitations and areas for future research

This study possesses several limitations that warrant consideration. Notably, the potential influence of visual stimuli on microsaccade rate alteration should be noticed, given its significance alongside cognitive processes. The observed variations in microsaccade rates between TOIs related to coherence or spatial proximity and the lack of disparities in TOIs connected to signaling might predominantly stem from differences in the visual complexity of stimuli rather than the workings of the working memory system. Addressing this issue necessitates comprehensive and in-depth investigations in the future.

In addition, the sample of participants was composed entirely of males, which may restrict the generalizability of the findings across genders. Another factor that might have affected the results is possible session fatigue, as the experimental tasks required sustained attention over time. Both aspects should be taken into account when interpreting the outcomes.

Moreover, an overarching challenge arises from the multifaceted nature of some metrics, entailing the need for a deeper understanding of how eye movements can effectively reflect cognitive processes. Consequently, forthcoming research endeavors should diligently explore this intricate relationship.

Furthermore, it is recommended that future investigations account for individuals' distinct learning styles, thereby enhancing the comprehensiveness of findings. Lastly, to attain a more comprehensive understanding of multimedia

learning dynamics, future studies should investigate the various forms of cognitive load (intrinsic, extraneous, and germane) and employ eye tracker metrics to measure them individually. By undertaking these proposed avenues of inquiry, multimedia learning can advance toward a more nuanced comprehension of its underlying mechanisms.

## Author contributions

**Conceptualization:** S. Hamid Amiri, Alireza Bosaghzadeh, Reza Ebrahimpour.

**Data curation:** Fatemeh Shahnabati, Atefeh Sabourifard.

**Formal analysis:** Fatemeh Shahnabati, Atefeh Sabourifard.

**Funding acquisition:** Reza Ebrahimpour.

**Investigation:** S. Hamid Amiri, Reza Ebrahimpour.

**Methodology:** Fatemeh Shahnabati, Reza Ebrahimpour.

**Project administration:** S. Hamid Amiri, Alireza Bosaghzadeh, Reza Ebrahimpour.

**Software:** Fatemeh Shahnabati, Atefeh Sabourifard.

**Supervision:** S. Hamid Amiri, Alireza Bosaghzadeh.

**Validation:** Fatemeh Shahnabati, Reza Ebrahimpour.

**Visualization:** Fatemeh Shahnabati, Reza Ebrahimpour.

**Writing – original draft:** Fatemeh Shahnabati, Atefeh Sabourifard.

**Writing – review & editing:** S. Hamid Amiri, Alireza Bosaghzadeh, Reza Ebrahimpour.

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
