## [Decision Letter · Decision Letter 0]

10 Jul 2025

Dear Dr. Ebrahimpour,

**Language Confound** : Reviewer 2 raised concerns about using English materials for Persian-speaking participants, which may have introduced extraneous cognitive load. Please discuss this limitation more fully.

**Statistical Rigor** : Given the small sample size, consider using non-parametric tests and add visual assessments of normality. Include an a priori power analysis.

**Figure Quality** : Improve figure resolution (≥300 dpi) and enhance AOI labeling for clarity.

**Limitations** : Add a brief discussion of other limitations, such as the all-male sample and session fatigue.

publication criteria  and not, for example, on novelty or perceived impact.

We look forward to receiving your revised manuscript.

Kind regards,

Vishal Bharmauria

Academic Editor

PLOS ONE

Additional Editor Comments (if provided):

Reviewers' comments:

Reviewer's Responses to Questions

**Comments to the Author**

1. Is the manuscript technically sound, and do the data support the conclusions?

Reviewer #1: Yes

Reviewer #2: Yes

2. Has the statistical analysis been performed appropriately and rigorously?

Reviewer #1: Yes

Reviewer #2: No

3. Have the authors made all data underlying the findings in their manuscript fully available?

Reviewer #1: No

Reviewer #2: No

4. Is the manuscript presented in an intelligible fashion and written in standard English?

Reviewer #1: Yes

Reviewer #2: Yes

Reviewer #1: I already reviewed a previous version of this manuscript. The revised manuscript is much improved. The authors have addressed the major concerns I raised previously, particularly around English language proficiency, the clarity and validity of their multimedia manipulations, and the use of eye-tracking measures. I appreciate the added detail in your methods and the more careful interpretation of the results, especially regarding fixation patterns, pupil size, and microsaccades.

A few minor points remain. Please consider improving the labeling and readability of AOIs in the scan path and fixation figures. Also, the overall quality of the figures is quite low. I’m not sure if that’s due to journal compression, but please ensure that the final version includes clear, high-resolution images.

Lastly, a brief mention of limitations (e.g., all-male sample, possible session fatigue) in the discussion would be helpful.

Reviewer #2: The paper titled "Cognitive Load and Visual Attention Assessment Using Physiological Eye-Tracking Measures in Multimedia Learning" presents an interesting investigation into how multimedia instructional design affects learners' cognitive load and visual attention patterns. The researchers employed eye-tracking technology to analyze ocular behaviors and their relationship with cognitive load and learning performance. While the study offers valuable insights, several critical issues need to be addressed before publication.

1- One notable limitation of this study is the use of English-language instructional materials for Persian-speaking participants, despite their native language being Persian. This issue may have influenced the results in the many ways. Particularly, it can increase the cognitive load. Even though participants had intermediate English proficiency, processing educational content in a non-native language requires additional cognitive effort. This could introduce extraneous cognitive load unrelated to multimedia design. Accordingly, some of the reported cognitive load differences between the two groups (P and NP) might stem from individual variations in English proficiency rather than the experimental manipulation. Also, although participants' language skills were assessed using an IELTS simulator test, individual differences in listening comprehension, processing speed, or vocabulary familiarity could affect their interaction with the content. For instance, participants with better comprehension of specific terms might experience lower cognitive load, even when exposed to the non-principled (NP) version. On the other hand, in can cause potential confounding effects on eye-tracking metrics and eye movements (e.g., fixation counts or durations) might reflect difficulty in understanding English text rather than multimedia design flaws. To address this limitation, I recommend using instructional materials in participants' native language to eliminate language-related cognitive load.

2- With 34 participants divided into two groups (approximately 14 per group after exclusions), the sample size in each group is relatively small. In such cases, normality tests like Kolmogorov-Smirnov or Shapiro-Wilk may have reduced sensitivity in detecting deviations from normality. Studies show these tests can produce false negatives with small samples (<30), potentially misidentifying non-normal data as normal. Instead, using non-parametric tests (e.g., Mann-Whitney U) could be preferable as they don't require normality assumptions. Visual methods like Q-Q plots or examining skewness/kurtosis could qualitatively assess normality (though not mentioned in the paper).

3- The lack of a priori power analysis represents a significant limitation because the sample size may be inadequate (i.e., the study might be underpowered to detect true effects). Also, the risk of false negatives (Type II errors) increases (where real significant differences are incorrectly deemed non-significant). Authors should report detailed power analysis (e.g., using G*Power) to justify sample sizes.

4- The resolution of the figures is currently insufficient for proper visualization. All figures must be rendered at a minimum of 300 dpi (dots per inch) to meet publication standards.

**Do you want your identity to be public for this peer review?** For information about this choice, including consent withdrawal, please see our Privacy Policy

Reviewer #1: No

Reviewer #2: No

---

## [Author Response · Author response to Decision Letter 1]

22 Sep 2025

Comments to the Author

We would like to express our sincere gratitude to the reviewers for their valuable, insightful, and constructive feedback on our manuscript. Your comments have been extremely helpful in refining the quality, clarity, and overall contribution of this work. We greatly appreciate the time and effort you devoted to a careful evaluation of our study.

We have carefully addressed each of the points raised and revised the manuscript accordingly. In the following pages, we provide a detailed, point-by-point response to all the comments and suggestions. We hope that the revisions made will meet your expectations and improve the manuscript to a level suitable for publication.

1. Is the manuscript technically sound, and do the data support the conclusions? The manuscript must describe a technically sound piece of scientific research with data that supports the conclusions. Experiments must have been conducted rigorously, with appropriate controls, replication, and sample sizes. The conclusions must be drawn appropriately based on the data presented.

Reviewer #1: Yes

Reviewer #2: Yes

We greatly appreciate your careful review and valuable feedback.

2. Has the statistical analysis been performed appropriately and rigorously?

Reviewer #1: Yes

Reviewer #2: No

We recognize the concerns raised by Reviewer #2. To address these issues, we have carefully considered all the points and suggestions you raised, and the necessary clarifications and revisions have been incorporated into the manuscript. Details of the changes are provided in the responses to the reviewer’s comments below.

3. Have the authors made all data underlying the findings in their manuscript fully available? The PLOS Data policy requires authors to make all data underlying the findings described in their manuscript fully available without restriction, with rare exception (please refer to the Data Availability Statement in the manuscript PDF file). The data should be provided as part of the manuscript or its supporting information, or deposited to a public repository. For example, in addition to summary statistics, the data points behind means, medians and variance measures should be available. If there are restrictions on publicly sharing data—e.g. participant privacy or use of data from a third party—those must be specified.

Reviewer #1: No

Reviewer #2: No

We have made the raw data and statistical analysis data available via a link mentioned in data availability statement for transparency and further verification.

4. Is the manuscript presented in an intelligible fashion and written in standard English? PLOS ONE does not copyedit accepted manuscripts, so the language in submitted articles must be clear, correct, and unambiguous. Any typographical or grammatical errors should be corrected at revision, so please note any specific errors here.

Reviewer #1: Yes

Reviewer #2: Yes

Reviewer #1:

I already reviewed a previous version of this manuscript. The revised manuscript is much improved. The authors have addressed the major concerns I raised previously, particularly around English language proficiency, the clarity and validity of their multimedia manipulations, and the use of eye-tracking measures. I appreciate the added detail in your methods and the more careful interpretation of the results, especially regarding fixation patterns, pupil size, and microsaccades.

A few minor points remain. Please consider improving the labeling and readability of AOIs in the scan path and fixation figures. Also, the overall quality of the figures is quite low. I’m not sure if that’s due to journal compression, but please ensure that the final version includes clear, high-resolution images.

Lastly, a brief mention of limitations (e.g., all-male sample, possible session fatigue) in the discussion would be helpful.

We sincerely thank the reviewer for the valuable suggestions regarding the figures. In response, the AOIs in Figure 7 have been clearly highlighted using distinct colors to improve visibility. In Figure 8, both the AOIs and their corresponding labels have been made more explicit. Additionally, the overall quality of all figures has been enhanced, and each now meets a minimum resolution of 300 dpi to ensure clear readability.

The mentioned limitations, including the all-male sample and possible session fatigue, have been added to the manuscript.

Reviewer #2:

The paper titled "Cognitive Load and Visual Attention Assessment Using Physiological Eye-Tracking Measures in Multimedia Learning" presents an interesting investigation into how multimedia instructional design affects learners' cognitive load and visual attention patterns. The researchers employed eye-tracking technology to analyze ocular behaviors and their relationship with cognitive load and learning performance. While the study offers valuable insights, several critical issues need to be addressed

before publication.

1- One notable limitation of this study is the use of English-language instructional materials for Persian-speaking participants, despite their native language being Persian. This issue may have influenced the results in the many ways. Particularly, it can increase the cognitive load. Even though participants had intermediate English proficiency, processing educational content in a non-native language requires additional cognitive effort. This could introduce extraneous cognitive load unrelated to multimedia design. Accordingly, some of the reported cognitive load differences between the two groups (P and NP) might stem from individual variations in English proficiency rather than the experimental manipulation. Also, although participants' language skills were assessed using an IELTS simulator test, individual differences in listening comprehension, processing speed, or vocabulary familiarity could affect their interaction with the content. For instance, participants with better comprehension of specific terms might experience lower cognitive load, even when exposed to the non-principled (NP) version. On the other hand, in can cause potential confounding effects on eye-tracking metrics and eye movements (e.g., fixation counts or durations) might reflect difficulty in understanding English text rather than multimedia design flaws. To address this limitation, I recommend using instructional materials in participants' native language to eliminate language-related cognitive load.

As previously mentioned, prior to the experiment all participants completed the standardized IELTS test, and their scores ranged between 6-7. This indicates that they all possessed a relatively high and comparable level of English proficiency. Participants were then randomly assigned to the P and NP conditions, which means that exposure to the second language was a common factor across both groups. Since there were no significant differences in language proficiency between the two conditions, it can be assumed that, on average, a similar level of cognitive load stemming from second language processing was imposed on both groups.

If the observed eye-movement behaviors and cognitive load were entirely attributable to the use of a second language, then no meaningful differences should have emerged between the P and NP conditions in terms of cognitive load, performance, or related measures. Furthermore, the instructional material employed was a standardized resource specifically designed for non-native English learners, which ensured a high level of control. The material is typically aimed at learners with proficiency levels equivalent to IELTS scores of approximately 5–6.5. Given that our participants’ proficiency scores ranged from 6-7, the vocabulary and content were appropriate for their level and unlikely to have posed substantial difficulty, as most participants were already familiar with the majority of the lexical items.

2- With 34 participants divided into two groups (approximately 14 per group after exclusions), the sample size in each group is relatively small. In such cases, normality tests like Kolmogorov-Smirnov or Shapiro-Wilk may have reduced sensitivity in detecting deviations from normality. Studies show these tests can produce false negatives with small samples (<30), potentially misidentifying non-normal data as normal. Instead, using non parametric tests (e.g., Mann-Whitney U) could be preferable as they don't require normality assumptions. Visual methods like Q-Q plots or examining skewness/kurtosis could qualitatively assess normality (though not mentioned in the paper).

To compare the two conditions, the non-parametric Mann–Whitney U test was applied. For the NASA data, the test yielded U = 74, p < 0.001, indicating a statistically significant difference between the conditions. Similarly, for the Recall data, U = 674.5, p < 0.001, also indicating a significant difference.

To assess the normality of the data distribution visually, Q-Q (Quantile-Quantile) plots were generated. The points closely followed the reference line, suggesting no substantial deviation from normality. For the NASA data, skewness (0.45) and kurtosis (0.37) fell within the acceptable range (−1 to +1), while for the Recall data, skewness (−0.47) and kurtosis (0.6) were also within this range. These results indicate approximate symmetry and a mesokurtic distribution.

The relevant descriptions have been added to the manuscript.

Below, you can see the corresponding Q-Q plot charts for the NASA-TLX and recall scores.

NASA-TLX:

Recall:

3- The lack of a priori power analysis represents a significant limitation because the sample size may be inadequate (i.e., the study might be underpowered to detect true effects). Also, the risk of false negatives (Type II errors) increases (where real significant differences are incorrectly deemed non-significant). Authors should report detailed power analysis (e.g., using G*Power) to justify sample sizes.

As you can see below, a priori power analysis was performed using G*Power to estimate the required sample size for the planned comparison. The analysis specified a two-tailed independent-samples t-test with an expected effect size of d= 0.7, a significance level of α=0.05, and desired statistical power of 1-β = 0.8 . The calculation indicated that 34 participants per group, corresponding to a total sample size of 68, would be sufficient to achieve the targeted power.

we believe it is reasonable to treat the two sessions per participant as independent data points, allowing us to double the total sample size from 34 individuals to 68 observations. Although each participant took part in two experimental sessions, we argue that these should be treated as independent observations. Each session was conducted on a different day and involved completely different video content. The participants were not exposed to the same material in both sessions. In one session, they were assigned to P condition, and in the other to NP condition. There was no overlap in content, no learning or transfer effects were expected, and the sessions were separated in time to minimize any carryover influence. Moreover, the order of conditions was randomized across participants to control for any potential order effects. That is, some participants started with P version, while others began with NP version. This consideration is also supported by the independence of stimuli and the randomization procedures applied.

4- The resolution of the figures is currently insufficient for proper visualization. All figures must be rendered at a minimum of 300

dpi (dots per inch) to meet publication standards.

We sincerely thank the reviewer for the valuable suggestions regarding the figures. The overall quality of all figures has been enhanced, and each now meets a minimum resolution of 300 dpi to ensure clear readability.

---

## [Decision Letter · Decision Letter 1]

6 Nov 2025

Cognitive load and visual attention assessment using physiological eye tracking measures in multimedia learning

PONE-D-25-22353R1

Dear Dr. Ebrahimpour,

We’re pleased to inform you that your manuscript has been judged scientifically suitable for publication and will be formally accepted for publication once it meets all outstanding technical requirements.

Kind regards,

Vishal Bharmauria

Academic Editor

PLOS ONE

Additional Editor Comments (optional):

Reviewers' comments:

Reviewer's Responses to Questions

**Comments to the Author**

Reviewer #1: All comments have been addressed

Reviewer #2: All comments have been addressed

2. Is the manuscript technically sound, and do the data support the conclusions?

Reviewer #1: Yes

Reviewer #2: Yes

3. Has the statistical analysis been performed appropriately and rigorously?

Reviewer #1: Yes

Reviewer #2: Yes

4. Have the authors made all data underlying the findings in their manuscript fully available?

Reviewer #1: (No Response)

Reviewer #2: No

5. Is the manuscript presented in an intelligible fashion and written in standard English?

Reviewer #1: Yes

Reviewer #2: Yes

Reviewer #1: I don't have any more comments. The authors have sufficiently responded to all of my concerns. The revised version is ready for publication.

Reviewer #2: (No Response)

**Do you want your identity to be public for this peer review?** For information about this choice, including consent withdrawal, please see our Privacy Policy

Reviewer #1: No

Reviewer #2: **Yes: ** Amirhossein Ghaderi

---

## [Editor Report · Acceptance letter]

PONE-D-25-22353R1

PLOS ONE

Dear Dr. Ebrahimpour,

I'm pleased to inform you that your manuscript has been deemed suitable for publication in PLOS ONE. Congratulations! Your manuscript is now being handed over to our production team.

Kind regards,

on behalf of

Dr. Vishal Bharmauria

Academic Editor

PLOS ONE